# A Study on High Strength, High Plasticity, Non-Heat Treated Die-Cast Aluminum Alloy

**DOI:** 10.3390/ma15010295

**Published:** 2021-12-31

**Authors:** Ruizhang Hu, Chun Guo, Mingliang Ma

**Affiliations:** 1College of mechanical engineering, Anhui Science and Technology University, Bengbu 233000, China; hurz@ahstu.edu.cn; 2Luoyang Ship Material Research Institute, Luoyang 471000, China; mml111111@126.com

**Keywords:** aluminum alloy, die casting, microstructure, properties, non-heat-treated

## Abstract

The non-heat-treated, die-cast aluminum alloy samples were prepared meticulously via die-casting technology. The crystal structure, microstructure, and phase composition of the samples were comprehensively studied through electron backscatter diffraction (EBSD), metallographic microscopy, spectrometer, and transmission electron microscopy (TEM). The microhardness and tensile properties of the samples were tested. The die-cast samples were found to have desirable properties by studying the structure and performance of the samples. There were no defects, such as pores, cold partitions, or surface cracks, found. The metallographic structure of the samples was mainly α-Al, and various phases were distributed at the grain boundaries. Before heat treating, α-Al grains were mainly equiaxed with a great number of second phase particles at the grain boundaries. After heat treating, the α-Al grains were massive and coarsened, and the second phase grains were refined and uniformly distributed, compared with those before the heat treating. The EBSD results showed that the grain boundary Si particles were solid solution decomposed after heat treatment. The particles became smaller, and their distribution was more uniform. Transmission electron microscopy found that there were nano-scale Al-Mn, Al-Cu, and Cu phases dispersed in the samples. The average microhardness of the samples before heat treating was 114 HV_0.1_, while, after the heat treating, the microhardness reached 121 HV_0.1_. The mechanical features of the samples were tremendous, and the obtained die-cast aluminum alloy had non-heat-treatment performance, which was greater than the ordinary die-cast aluminum alloys with a similar composition. The tensile strength of the aluminum alloys reached up to 310 MPa before heat treatment.

## 1. Introduction

Aluminum and aluminum-based alloys possess many appropriate properties, including great specific strengths, low densities, good plasticity, excellent mechanical characteristics, low thermal expansion coefficients, good resistance against corrosion in different media, notable electrical/thermal conductivities, easy processing, and recyclability [1,2,3,4]. Aluminum and its alloys have been looked at for a plethora of applications in the chemicals industry, aerospace infrastructure, aviation, packaging, automobile, shipbuilding, and machinery manufacturing. Very recently, this non-ferrous metal has played a key role in modern industries, such as the automobile industry [5,6]. The densities of aluminum alloys are estimated to be around 1/3 that of steel. According to reports, when high-strength steel, mild steel, or cast iron was replaced with an aluminum alloy, a weight reduction of 30 to 60% was achieved. Moreover, the utilization of aluminum alloys per kilogram could be reduced by 13–20 kg of greenhouse gas emissions [7,8]. Substituting aluminum for steel has become a development trend in lightweight automotive technology, and its use in luxury cars has been more obvious. Aluminum alloys for automobiles mainly include deformed and cast aluminum alloys, of which cast aluminum alloys dominate, accounting for about 80% of automotive aluminum. They have been mainly used to manufacture engine cylinder blocks, cylinder heads, clutch shells, bumpers, and wheels. Deformed aluminum alloys have been mainly used in the manufacture of body panels, such as the Audi A8 all-aluminum body. In addition, aluminum-based composite materials, foamed aluminum, and powder metallurgy aluminum alloys have also been used in automobiles. For example, Al-Si binary near-eutectic aluminum-silicon alloy was suitable for manufacturing complex, thin-walled, and non-strict load-bearing parts, such as instrument panels, guard plates, and housings [9,10]. Therefore, if a large number of aluminum alloys can be used to replace some steel parts and materials, the weight of the product can be greatly reduced while ensuring its performance, which will greatly reduce energy consumption, save resources, and also reduce emissions [11,12,13]. However, the long-term problems that have plagued these alloys are their low strengths and poor machining performances, which limits their use. At the same time, with the development of the automobile manufacturing industry, higher requirements have been put forward for the properties of the materials used. The development of new types of aluminum alloys with high strengths, high plasticity, high toughness, and good processing and manufacturing properties has important research significance.

Currently, most of the aluminum-silicon materials on the market have disadvantages, such as poor mechanical properties and poor cutting performance. In addition, most of the aluminum alloy preparation has been completed by heat treatment, which consumes huge costs. This puts higher requirements on the alloy materials and their preparation processes.

Non-heat-treated aluminum alloys are a new type of aluminum alloy material that has emerged in recent years. Its characteristics were that the parts do not need to undergo high-temperature solution treatment and artificial aging. Only natural aging of aluminum alloys can achieve high strengths and plasticity to satisfy the performance requirements for automobile and aerospace parts [14,15]. After a literature review, it was found that there were only a few research reports on non-heat-treated aluminum alloys, among which there were few reports on high-strength, high-plasticity, non-heat-treated aluminum alloys.

Therefore, this study used die-casting technology to create high-strength, high plasticity, non-heat-treated die-cast aluminum alloy. The crystal structure, microstructure, phase composition, and mechanical properties of the samples were comprehensively studied. This investigation presents data to enhance the application of high-strength, high plasticity, non-heat-treated die-cast aluminum alloys.

## 2. Materials and Methods

The experimental raw materials were 99.70% (m/m) industrial pure aluminum, 99.95% (m/m) pure magnesium, and Al-8Si, Al-10Mn, Al-40Cu, Al-10Ti, Al-20Fe, Al-10Cr, and Al-10Sr master alloys. The alloy was smelted in a crucible resistance furnace. The smelting process started by adding Al-8Si, Al-10Mn, Al-10Ti, Al-20Fe, Al-10Cr, and Al-10Sr master alloys to the remelted pure aluminum ingot when the furnace was heated at 730 °C. After all of the metals had melted, the molten metal was kept at 740 °C for 30 min and stirred for 8 min. The pure Mg wrapped in aluminum foil was pressed into the molten aluminum with a bell when the temperature of the molten metal was reduced to 700 °C. Once the metals were melted, they were stirred for 10 min, then the temperature was raised to 720 °C for 30 min, and slag was removed. Then, the molten metal was cooled to 690 °C for 30 min to allow for any degassing. The molten metal was poured into molds by a cold chamber horizontal die casting machine (LH400T Bengbu Longhua Die Casting Machines Co., Ltd. Bengbu, China). The metal molds with painting were preheated to 180 °C before die casting, and the cast samples were obtained. A T6 treatment (heating to 505 °C for 1 h, then holding for 8 h, and quenching, followed by heating to 175 °C for 1 h, holding for 6 h, followed by cooling in the furnace) was done using an XL-1200 box-type electric furnace and JSL-1000 pit furnace (SiYang Piecision equipment(shanghai)Co.,Ltd.Shanghai, China).

An example of the die-cast samples in this study is shown in Figure 1. The testing samples were cut from the middle part of the sample for metallurgical studies (SEM and hardness tests), chemical composition analysis, and transmission electron microscopy (TEM). Analysis of the chemical composition was performed using GB/T 4336-2016, via QSN750-II direct considering the spectrometer. The tensile strengths of the samples were measured following the standard GB/T228.1-2010. The test of tensile behavior was performed at room temperature by using a 100 kN material testing technique (SINTECH20/G). Furthermore, the displacement was measured during the stretching process, and the initial strain rate was considered 0.005 min^−1^. The metallographic and SEM samples were prepared by cutting, polishing, and etching with a picric acid alcohol solution. The samples were observed using the OLYMPUS GX71 metallographic microscope (Olympus Corporation, Tokyo, Japan) according to the standard GB/T 13298-2015 “Metal Microstructure Inspection Method.” The samples under great magnification were monitored through the Quanta600 scanning electron microscope (FEI, Hillsboro, OR, USA). For the TEM samples, a 0.5 mm, thin piece was cut from the block sample and was meticulously ground to a thickness of around 120 μm. Afterward, three discs were punched out from the sample and were ground to thicknesses of about 50 μm. The studied perforated film samples were attained via double spray electrolysis using a Gatan691 ion thinner at −20 °C and 75 V, with a 4% perchloric acid in alcohol solution over 1.0 h. The crystal structures of the samples were observed by a JEM-2100 model transmission electron microscope (TEM) with an accelerating voltage of about 200 kV. Regarding the GB/T 4340.1-2009 “Metallic material Vickers hardness test Part 1: Test method” standard, the micro-Vickers hardness measurement was performed using a VMH-104 with a test force of 100 g and a pressure holding time of 10 s. The average value was taken from 10 measured points. Table 1 shows the chemical composition of the die-cast samples.

## 3. Results

### 3.1. Forming Characteristics of Non-Heat-Treated, Die-Cast Aluminum Alloy

Figure 1 shows a non-heat-treated, die-cast aluminum alloy sample. The sample was well-formed without defects, such as pores, cold barriers, or surface cracks. This indicated that the non-heat-treated, die-cast aluminum alloy designed in this research had a good die-casting performance.

### 3.2. Non-Heat-Treated, Die-Cast Aluminum Alloy Composition and Structural Characteristics

The chemical composition of the non-heat-treated, die-cast aluminum alloy samples is shown in Table 1. The main alloying elements of the die-cast samples were Si, Cu, Mn, Mg, and Fe. The contents were 8.55%, 2.50%, 0.660%, 0.326%, and 0.208%, respectively. The trace alloying elements were Ti, Cr, and Sr with 0.091%, 0.088%, and 0.02%, respectively. Si was the main alloying element and was the main component to improve the fluidity of the alloy. In addition, Si improved the tensile strength, hardness, machinability, and strength at high temperatures, while reducing elongation [16,17,18]. To improve various characteristics of aluminum alloys, such as corrosion resistance and mechanical strength, copper (Cu) was mainly utilized within these alloys. Aluminum-copper alloys have a face-centered cubic crystal structure, so this type of alloy has good ductility and better machinability [19,20,21,22,23]. Manganese (Mn) prohibited recrystallization within the structure of the aluminum alloy, raised the temperature of recrystallization, and refined the recrystallization grains considerably. Mn was an essential element of aluminum-based alloys. Al-Mn binary alloys could be incorporated individually, although combined along with other elements. Thus, most aluminum alloys included Mn [24,25]. Mg was mainly used in aluminum alloys to increase tensile strength, hardness, and corrosion resistance. Mg can improve the performance of the anodic oxide film, increase the strength and yield limit, and improve the machinability of the alloy by adding a small amount (about 0.2 to 0.3%) to high-silicon aluminum alloys. The aluminum alloy containing 8 % Mg had excellent corrosion resistance [26,27,28,29,30]. The main beneficial effect of Fe in aluminum alloys was to reduce sticking to the mold. A certain amount of iron in the alloy facilitated mold ejection. The strength and hardness of the alloy can be improved, and the tendency of hot cracking can be reduced, when the iron content was less than 1.7%. In Al-Si alloys, iron existed in two common phases: α-AlFeSi and β-AlFeSi. The α-AlFeSi increased the strength and hardness without much reduction in toughness, while the needle-shaped β-AlFeSi phase split the matrix, significantly reducing the toughness of the alloy, especially the impact toughness [31,32,33]. Cr played a role in the dispersion strengthening of the alloy. Cr formed (CrMn) Al12 and (CrFe) Al7 intermetallic compounds within the aluminum structure, inhibiting the nucleation and growth attributed to recrystallization. Cr strengthened the alloy to increase the mechanical features of the cast aluminum alloy [34,35,36]. A small amount of titanium (Ti) in the alloy enhanced the mechanical properties. The alloying Ti delayed the formation of metastable phases in the alloy and the over-aging of the alloy. During the aging process, Ti and other elements formed a dispersed TiAl3 phase and an Al-Si-Cu-Ti quaternary compound phase which strengthened the dispersion in the alloy matrix [37,38]. Strontium (Sr) was a surface-active element. In crystallography, Sr changed the behavior of the intermetallic compound phase. Therefore, modification with Sr improved the plastic workability of the alloys and the quality of the final products. Due to the advantages of long modification time, good effect, and reproducibility, Sr has replaced Na within Al-Si casting alloys in the past few years. Next, 0.015–0.03% Sr was added to the Al alloy for extrusion, and can make the β-AlFeSi phase turn into the α-AlFeSi phase, to improve the mechanical features of the material and the surface roughness related to the product. Adding Sr to the hypereutectic AlSi alloy led to a reduction in the size of the primary silicon particles and boosted the plastic processing performance. It can be smoothly hot rolled or cold rolled.

Figure 2 showed the XRD patterns of the non-heat-treated, die-cast samples before and after heat treating. According to Figure 2, the main phase composition of the sample was Al and Si. The main phases of the die-cast samples had not changed significantly by comparing the XRD patterns before and after the heat treatment.

The metallographic structure photos of the non-heat-treated, die-cast samples before and after heat treatment are shown in Figure 3. The metallographic structure of the samples before heat treatment was mainly α-Al with a dendritic network distribution of various phases shown in Figure 3a. The metallographic structure of the sample after heat treatment was mainly α-Al with uniformly distributed phases in Figure 3b. The α-Al grains before heat treatment were mainly equiaxed with a large number of second phase particles at the grain boundaries. After heat treatment, the α-Al grains were mainly in noticeable, coarse lumps. The second phase grains were refined and uniformly distributed compared with the second phase grains at the grain boundaries. After a solution heat treatment, the coarse, second-phase structure was dissolved and uniform, so the grain boundary interval was smaller compared with Figure 3a. In addition, no clear casting defects were seen, for instance, cracks or shrinkage holes, from the high-magnification metallographic photos (Figure 3). This indicated that the internal quality of the samples was good.

The EBSD phase distribution maps of the non-heat-treated, die-cast samples before and after heat treatment are shown in Figure 4. Both Al and Si in the die-cast sample had a face-centered cubic structures. The Si grain boundary particles were found to be a solid solution that had decomposed after heat treatment by comparing the EBSD phase distribution maps of the samples before and after heat treatment. The particles became smaller and the distribution was more uniform, which was consistent with the results obtained from the metallographic structure picture.

Figure 5 and Figure 6 presented the EBSD maps, pole maps, and reverse pole maps in different directions of the non-heat-treated, die-cast samples before and after heat treatment. Before the heat treatment, Al tended to orient along the (001) crystal orientation in the X0 direction, and Si tended to orient along the (111) crystal orientation. There was no obvious crystal orientation in the Y0 and Z0 directions, as shown in Figure 5. After heat treatment, Al tended towards the (001) crystal orientation in the X0 direction, with no significant change from before the heat treatment, shown in Figure 6. However, the crystal orientation of Si in the X0 direction made a significant change, moving to the (101) and (001) crystal facets from the original (111) orientation. This proved that Si had undergone solid solution decomposition during the heat treatment. This agreed with the results of the metallographic structure pictures and the EBSD phase distribution maps.

To gain a deeper understanding of the crystal structure of the non-heat-treated, die-cast samples, Figure 7 and Figure 8 show TEM images of the samples. The crystal structures of the sample before and after heat treatment presented a massive grain structure when observed on a high magnification transmission electron microscope (TEM). The electron diffraction pattern analysis results of the selected study area showed that the matrix was Al phase, which was consistent with the XRD results. However, a large number of dispersed second phases with small sizes, down to the nanoscale, appeared in the Al matrix after heat treatment, as seen by comparing Figure 7a,b. This was the main reason for the increased tensile strength. TEM images at different positions of the sample before heat treatment are shown in Figure 8. Figure 8a shows that the micron-sized, large particle reinforcement phase, which was determined to be Si by EDS, was distributed on the grain boundary of Al. The results were consistent with the EBSD phase distribution diagram. The existence of Al-Mn and Al-Cu phases was found in Figure 8b,c. The existence of dispersed nano-Cu phases was detected by a high-resolution lattice diagram, as shown in Figure 8d.

### 3.3. Mechanical Properties of Non-Heat-Treated, Die-Cast Aluminum Alloy Samples

The microhardness test results of the non-heat-treated, die-cast samples before and after heat treatment are shown in Table 2. From the microhardness test results, the hardness distribution of the samples before heat treating was between 108–122 HV_0.1_, and the average value was 114 HV_0.1_. The hardness distribution range after heat treatment was 111–128 HV_0.1_, and the average value was 121 HV_0.1_. Overall, the microhardness of the samples increased slightly after heat treatment. The dispersion of the second phase after heat treatment was responsible for these increases. On the whole, the hardness distribution of the samples was relatively uniform, and the fluctuations in the microhardness were due to the modification of the structure.

The results of investigating the mechanical features of the non-heat-treated, die-cast samples are shown in Table 3. The mechanical features of the samples were excellent. The yield strength before heat treating was 158 MPa, the tensile strength was 310 MPa, and the elongation was 6.9%. After heat treating, the yield strength was 289 MPa, the tensile strength was 375 MPa, and the elongation was only 4.3%. It was worth noting that the tensile strength of the die-cast aluminum alloy obtained in this study before heat treatment reached up to 310 MPa, which was greater than the strength of the ordinary cast aluminum alloy with a similar composition (ASTM B 179-06, alloy code YL104, tensile strength 220 MPa, Elongation rate 2%) (see Table 4). This indicated that the die-cast aluminum alloy obtained in this study had non-heat treatment properties. The second phase (such as Al-Mn, Al-Cu, and Cu) in the sample was the main reason for the high tensile strength, according to the results obtained from the metallographic diagram, EBSD maps, and transmission electron microscope pictures of the samples before heat treatment. In addition, the die-cast aluminum alloy obtained in this study can reach a higher strength of 375 MPa after the T6 heat treatment.

To analyze the mechanism of the fracture related to the samples, Figure 9 shows the fracture morphology of the non-heat-treated, die-cast samples after testing. The tensile fracture of the sample before heat treatment was dimple-like, indicating that the plasticity of the sample was better, which was consistent with the larger elongation of the tensile sample.

## 4. Conclusions

The non-heat-treated aluminum alloy samples were prepared using die-casting technology. The samples prepared by die-casting had good forming quality without defects, such as pores, cold barriers, or surface cracks. The metallographic structure of the samples was mainly α-Al and various phases distributed in the grain boundaries. Before heat treatment, α-Al grains were mainly equiaxed with a great number of second-phase particles at the grain boundary. The latter α-Al grains were massive and coarsened, and the second phase grains were refined and uniformly distributed compared with those before the heat treatment. The EBSD results showed that the grain boundary Si particles were a decomposed solid solution after heat treatment. The particles became smaller and the distribution was more uniform. Transmission electron microscopy showed that there were nanoscale dispersed Al-Mn, Al-Cu, and Cu phases in the sample.

The average microhardness of the sample before heat treatment was 114 HV_0.1_, and after heat treatment was 121 HV_0.1_. The mechanical characteristics of the samples were excellent, and the obtained die-cast aluminum alloy had non-heat treatment, self-strengthening properties. The yield strength before heat treating was 158 MPa, the tensile strength was 310 MPa, and the elongation was 6.9%. After heat treatment, the yield strength was 289 MPa, the tensile strength was 375 MPa, and the elongation was 4.3%.

The alloy can be used in automotive, mechanical, rail transit, and other fields to replace lightweight body structures designed with sheet metal or die-cast structures related to collision performance in automobiles. This investigation provided useful data to enhance the application of high-strength, high plasticity, non-heat-treated, die-cast aluminum alloys.

## Figures and Tables

**Figure 1 materials-15-00295-f001:**
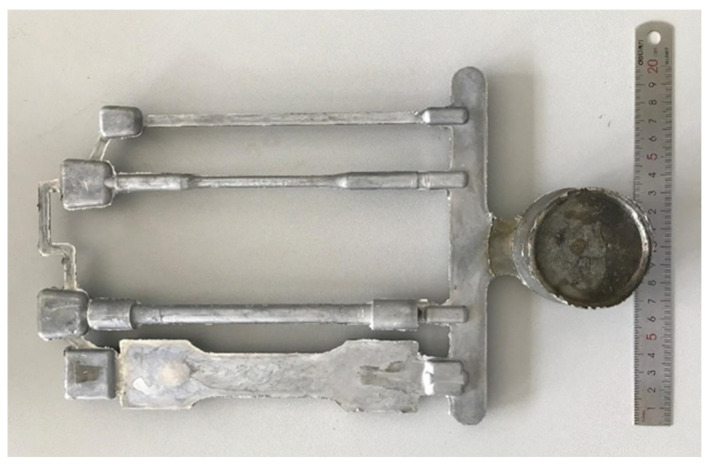
Photograph of a non-heat-treated, die-cast aluminum alloy sample.

**Figure 2 materials-15-00295-f002:**
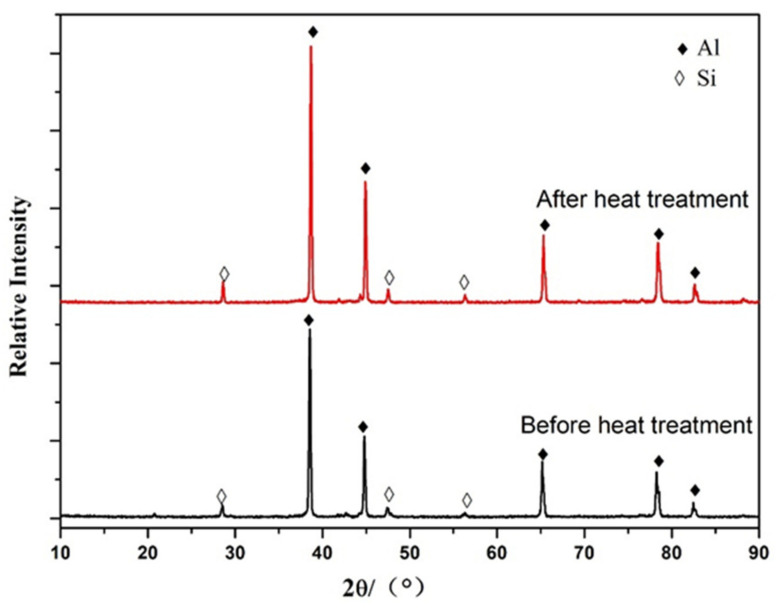
XRD patterns of non-heat-treated, die-cast aluminum alloy samples before and after heat treatment.

**Figure 3 materials-15-00295-f003:**
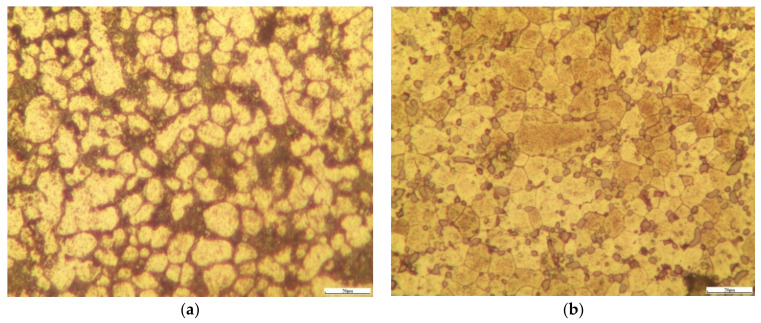
Metallographic structure of a non-heat-treated, die-cast sample: (**a**) before heat treating; (**b**) after heat treating.

**Figure 4 materials-15-00295-f004:**
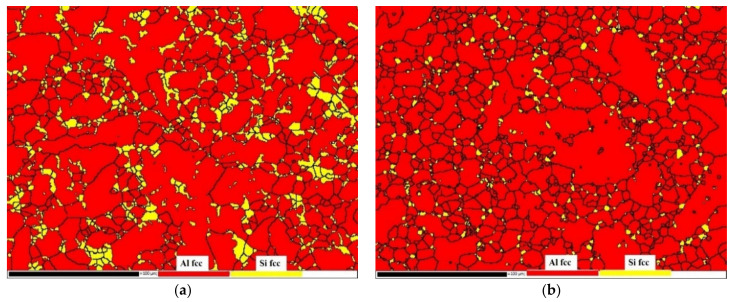
EBSD maps of non-heat-treated, die-cast samples: (**a**) before heat treatment; (**b**) after heat treatment.

**Figure 5 materials-15-00295-f005:**
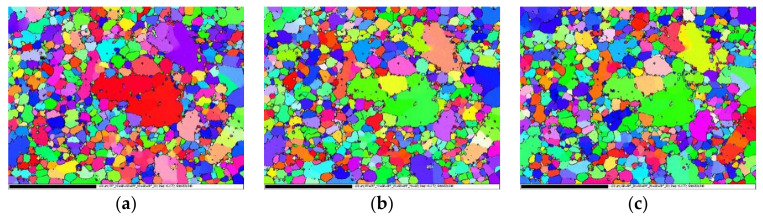
EBSD maps of non-heat-treated, die-cast samples before heat treatment: (**a**) X0 direction, (**b**) Y0 direction, (**c**) Z0 direction, (**d**) Al pole figures, (**e**) Si pole figures, (**f**) Al reverse pole figures, (**g**) Si reverse pole figures.

**Figure 6 materials-15-00295-f006:**
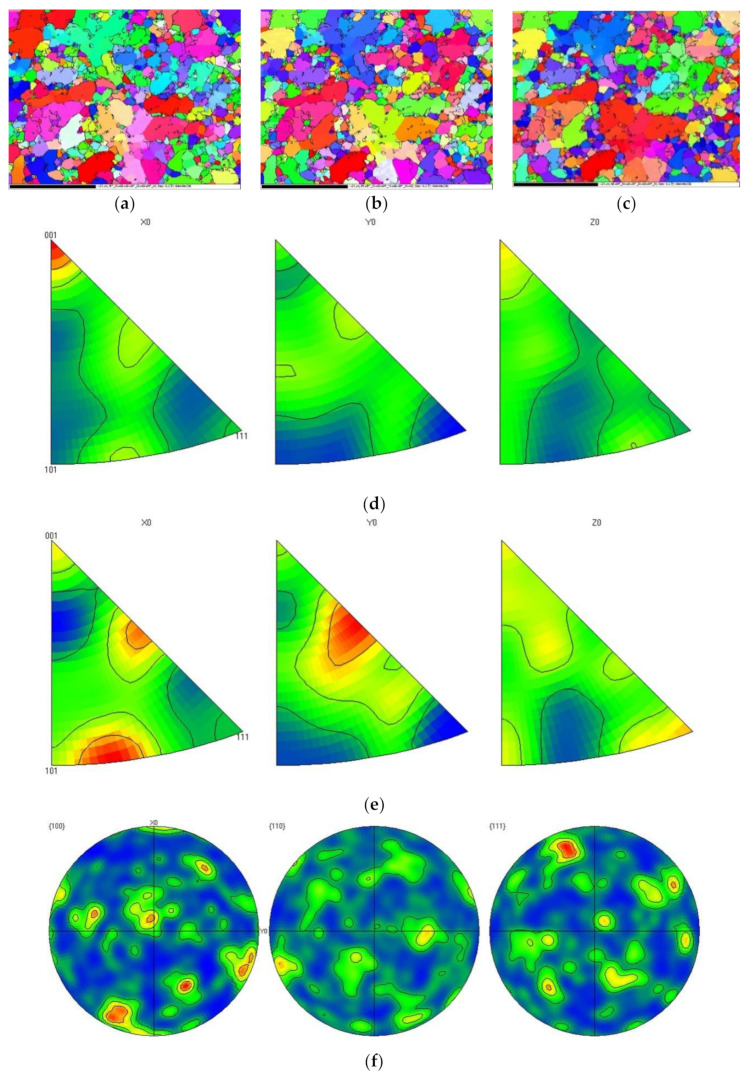
EBSD maps of non-heat-treated, die-cast samples after heat treatment: (**a**) X0 direction, (**b**) Y0 direction, (**c**) Z0 direction, (**d**) Al pole figures, (**e**) Si pole figures, (**f**) Al reverse pole figures, (**g**) Si reverse pole figures.

**Figure 7 materials-15-00295-f007:**
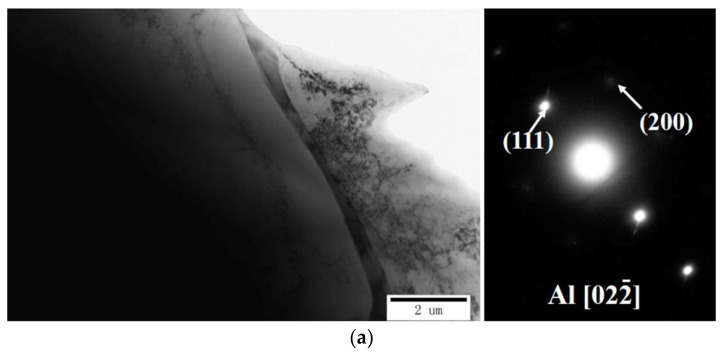
The crystal structure of the non-heat-treated, die-cast aluminum alloy sample: (**a**) before heat treatment; (**b**) after heat treatment.

**Figure 8 materials-15-00295-f008:**
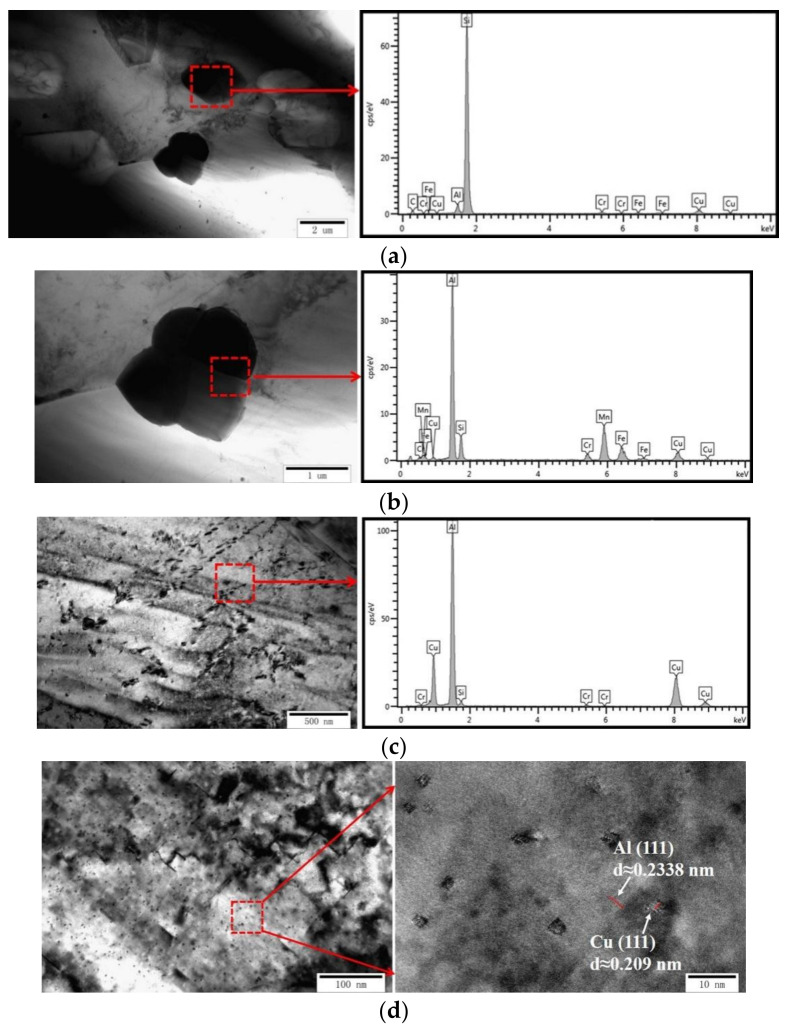
TEM images of the crystal structure of the non-heat-treated, die-cast aluminum alloy samples before heat treating: (**a**) Si phase, (**b**) Al-Mn phase, (**c**) Al-Cu phase, (**d**) high-resolution crystal grid.

**Figure 9 materials-15-00295-f009:**
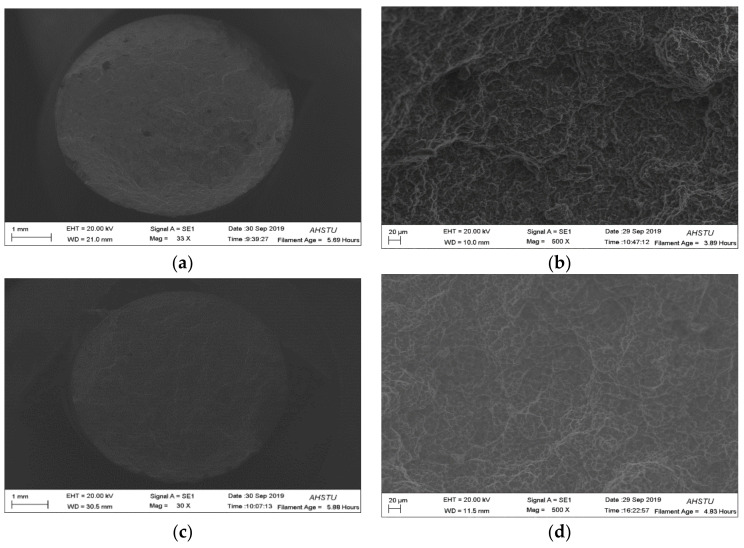
Impact fracture morphologies of the non-heat-treated die-cast aluminum alloy samples: (**a**) fracture surface before heat treating, (**b**) high magnification fracture surface before heat treating, (**c**) fracture surface after heat treating, (**d**) high magnification fracture surface after heat treating.

**Table 1 materials-15-00295-t001:** The chemical composition of the die-cast sample (mass fraction/%).

Element	Fe	Cu	Mg	Mn	Ti	Si	Cr	Sr	Al
Sample	0.208	2.50	0.326	0.660	0.091	8.55	0.088	0.02	Bal.

**Table 2 materials-15-00295-t002:** Microhardness of non-heat-treated die-casting samples.

Sample	1	2	3	4	5	AVG	Errors
Before heat treatment	108	122	112	111	118	114	5.7
After heat treatment	124	111	116	124	128	121	6.9

**Table 3 materials-15-00295-t003:** Mechanical properties of non-heat-treated, die-cast aluminum alloy samples.

Sample	Yield Strength/MPa	Tensile Strength/MPa	Elongation (%)
Before heat treatment	158 ± 6	310 ± 4	6.9 ± 0.4
After heat treatment	290 ± 10	375 ± 6	4.3 ± 0.5

**Table 4 materials-15-00295-t004:** Typical mechanical properties of die-cast aluminum alloy test samples.

Reference	Alloy	Yield Strength/MPa	Tensile Strength/MPa	Elongation (%)
[39]	YZAlSi10Mg	-	200	2.0
[39]	YZAlSi12	-	220	2.0
[39]	YZAlSi9Cu4	-	320	3.5
[39]	YZAlSi11Cu3	-	230	1.0
[39]	YZAlSi17Cu5Mg	-	220	1.0
[39]	YZAlMg5Si1	-	220	2.0

## Data Availability

The data presented in this study are available on request from the corresponding author.

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
