# Peer review of "A Study on High Strength, High Plasticity, Non-Heat Treated Die-Cast Aluminum Alloy"

_materials, 2021, doi:10.3390/ma15010295_

Round 1

Reviewer 1 Report

  1. In the introduction, it is unacceptable to write phrases such as “a literature search” because scientific society generally accepted to use “a literature review.” The article should be read by a native speaker to avoid such incidents.
  2. Zone axes must be indicated in the selected area electron diffractions.
  3. In Tables 2 and 3, the values of mechanical characteristics are shown without errors. Please, add it.

Author Response

Point 1: In the introduction, it is unacceptable to write phrases such as “a literature search” because scientific society generally accepted to use “a literature review.” The article should be read by a native speaker to avoid such incidents.

Response 1: Thank you for reminding us the improper description. We have replaced "a literature search" with the phrases "a literature review". We have carefully revised the entire paper to avoid spelling and grammatical errors. In addition, we found a professional language editing agency to revise and polish our paper (Editing certificate as follows).

Point 2: Zone axes must be indicated in the selected area electron diffractions.

Response 2: We would like to thank you for the useful suggestions. We have added zone axes in the selected area electron diffractions. (see figure 7)

Figure 7: The crystal structure of the non-heat-treated, die-cast aluminum alloy sample: (a) before heat treatment (b) after heat treatment.

Point 3: In Tables 2 and 3, the values of mechanical characteristics are shown without errors. Please, add it.

Response 3: Thanks for the reviewer's good suggestions. We have revised Tables 2 and 3 as suggested.

Reviewer 2 Report

The Review “A Study on High Strength, High Plasticity, Non-Heat Treated 2 Die-Cast Aluminum Alloy” is dedicated to widely researched Al-Si-based alloys. This study in detail investigated mechanical properties (Yield/Tensile Strength and Microhardness) and microstructure of non-heat-treated, die-cast Al-Si alloy. Unfortunately, there are some questions to the authors, which make it difficult to publish the article in the Materials journal.

1) Could you please clarify the novelty of this study? Usually, at the end of a literature review, the research goal is formulated. I recommend rewriting lines 74-77 into an article in the best way.

2) Figure 1 shows photograph of a non-heat-treated, die-cast aluminum alloy with different shaped samples: round and rectangular. Do properties compare on different-shaped samples in the article? How do the mechanical properties depend on the cross-section of the sample? If in the article has not studied this point, why did the authors show the photo?

3) In Figure 4,5,6 shows EBSD maps not “diagrams”.

Author Response

Point 1: The Review “A Study on High Strength, High Plasticity, Non-Heat Treated 2 Die-Cast Aluminum Alloy” is dedicated to widely researched Al-Si-based alloys. This study in detail investigated mechanical properties (Yield/Tensile Strength and Microhardness) and microstructure of non-heat-treated, die-cast Al-Si alloy. Unfortunately, there are some questions to the authors, which make it difficult to publish the article in the Materials journal.

 Response 1: We would like to express our gratitude for your comments, and thank you for giving us helpful suggestions. We have revised the paper according to your comments and suggestions.

Point 2: Could you please clarify the novelty of this study? Usually, at the end of a literature review, the research goal is formulated. I recommend rewriting lines 74-77 into an article in the best way.

Response 2: We would like to thank you for the useful suggestions.

Novelty of this study: Non-heat-treated aluminum alloys are a new type of aluminum alloy material that has emerged in recent years. Its characteristics were that the parts do not need to undergo high-temperature solution treatment and artificial aging. Therefore, this study used die-casting technology to create high-strength, high plasticity, non-heat-treated die-cast aluminum alloy. The crystal structure, microstructure, phase composition, and mechanical properties of the samples were comprehensively studied. The tensile strength of the aluminum alloys reached up to 310 MPa before heat treatment.The die-cast samples were found to have desirable properties by studying the structure and performance of the samples.

We have rewritten lines 74-77 according to the review comments.

Point 3: Figure 1 shows photograph of a non-heat-treated, die-cast aluminum alloy with different shaped samples: round and rectangular. Do properties compare on different-shaped samples in the article? How do the mechanical properties depend on the cross-section of the sample? If in the article has not studied this point, why did the authors show the photo?

Response 3: The die-casting mold we use can simultaneously die-cast samples of different shapes and sizes for testing of different properties. This article uses tensile samples.

Point 4: In Figure 4,5,6 shows EBSD maps not “diagrams”.

Response 4: Thank you for reminding us the improper description. The word “diarams” was changed to “maps”.

Round 2

Reviewer 2 Report

According to the revised results, I agree that the paper will be published in this journal.

Author Response

Reply to Academic Editor

Point 1: Reviewer #2 already asked for a detailed statement on novelty of the work presented.

Authors addressed this question in part.

However, more details should be provided at this point: Authors should include a table listing mechanical properties of conventional high-strength aluminum alloys for comparison.

This will significantly strengthen the authors findings.

 Response 1: Thanks for the editor's good suggestions. We have included a table listing mechanical properties of conventional high-strength aluminum alloys for comparison (see Table 4).
